# Post-marketing quality surveillance of selected antibacterial agents marketed in porous borders; the case of Ethiopia-Sudan-Eritrea border

Tewodros Denekew[1,2], Tadele Eticha[2], Yehualashet Teshome[3], Siyalkalem Endeshaw[4], Ayenew Ashenef[2,5]*

1 Department of Pharmaceutical Chemistry, College of Health Science, University of Gondar, Gondar, Ethiopia, 2 Department of Pharmaceutical Chemistry and Pharmacognosy, School of Pharmacy, College of Health Sciences, Addis Ababa University, Addis Ababa, Ethiopia, 3 Department of Pharmacy, Pharmaceutical Chemistry Unit, Asrat Woldeyes Health Science Campus, Debre Birhan University, Debre Birhan, Ethiopia, 4 Ethiopian Food and Drug and Authority (EFDA), Addis Ababa, Ethiopia, 5 Center for Innovative Drug Development and Therapeutic Trials for Africa (CDT-Africa), College of Health Sciences, Addis Ababa University, Addis Ababa, Ethiopia

* ayenew.ashenef@aau.edu.et

## Abstract

### Background

The presence of poor-quality medicines is becoming a public health threat in many parts of the world, particularly in developing countries. Antibiotics are among the most common anti-infective medicines that are highly prone to this problem.

### Objectives

The purpose of this study was to assess the quality of selected antibacterials that are marketed in **Setit Humera** and **West Gondar** Zones, North West Ethiopia, which are located on the Ethiopian side of the Ethiopia-Sudan-Eritrea border.

### Methods

Seventy-one samples of the four antibacterial medicines (Ciprofloxacin, Norfloxacin, Amoxycillin, and Amoxycillin clavullanate combination) were collected from six sites in **Setit Humera** and **West Gondar** Zones, North West Ethiopia. A mystery shopping system was used for sample collection. Visual inspections and confirmation of the registration status were carried out using the joint WHO/FIP/USP checklist and the Ethiopian Food and Drug Authority's (EFDA's) Electronic Regulatory Information System (eRIS), respectively. Then Pharmacopeial methods (USP, BP) were employed to assess the physicochemical quality parameters.

### Results

During the period of our data collection, the application status for registration in the eRIS system was checked. From 71 samples, 25.35% (18/71) were not registered, and 15.49%

**Funding:** Addis Ababa University to TD Ministry of Innovation and technology, government of Ethiopia to AA. The funders had no role in study design, data collection and analysis, decision to publish, or preparation of the manuscript.

**Competing interests:** The authors have declared that no competing interests exist.

(11/71) were registered, but the license period had expired. Some samples (12.06% (17/71)) did not meet the visual inspection criteria. The physicochemical evaluation showed that all the samples studied met the quality specifications for the identification and hardness tests. However, concerning assay, dissolution, uniformity of dosage units, disintegration, and friability test parameters, 27.49% (23/71), 16.9% (12/71), and 14.08% (10/71), 2.82% (2/71) and 8.57% (3/35) of samples were found to be substandard, respectively. Overall, 56.33% (40/71) of the samples tested were of poor quality, failing to meet any one or more of the assessed parameters in this study.

## Conclusion

The study indicated that poor-quality antibacterial medicines are circulating in the study sites. Therefore, even if the area was affected by conflict at the time of the study, the regulatory bodies should focus on enforcing the necessary measures by collaborating with the regional and national regulatory medicine agencies to ensure that the antibacterial medicines available meet the required mandatory minimum standards.

## Introduction

The term antibiotic was derived from the Greek word biotikos, which means "against life" [1, 2]. These classes of medicines are strong and effective life-saving medicines used for treating infectious diseases of bacterial, fungal, and, in some cases, parasitic origin [3]. In developing countries, antibiotics are among the most sub-standardized and falsified medicines [4]. Thus, persistent post-market quality evaluation helps to produce clear information on the current quality status of the different brands of a given drug in circulation. It helps to determine a biopharmaceutical and therapeutically equivalent list of the products for the prescribers and users [5].

The WHO estimates that approximately 10% of pharmaceutical supply amount in low- and middle-income countries (LMICs) is engulfed with substandard and falsified (SF) products, the value reaching up to 50% of the medicine supply system in some developing countries, while it is as low as 1% in the developed world [6]. The prevalence of poor-quality medicines is emerging as a public health threat in developing countries. It is also a cause for concern across the globe [6]. The use of poor -quality antibiotics increases the risk of emergence as well as the spread of antimicrobial -resistant (AMR) strains, one cause for treatment failure, leads to the loss of trust in healthcare systems, and other associated socioeconomic consequences [7]. Thus, the quality of marketed drug products cannot be guaranteed unless proper regulations that are supported by active post-marketing surveillance and subsequent proper regulatory actions and measures by national, regional, and international regulatory bodies are undertaken [8]. The high rate of infectious diseases and, thus the subsequent high morbidity rates are factors inciting traffickers of falsified medicines to sell their products in developing countries. Hence, Ethiopia might be one of the destination countries for falsified antibiotics [9].

Ethiopia, the second most populous African country located in East Africa, is one of the sub-Saharan African countries where the pharmaceutical sector is guided by a national drug policy [10]. Amoxicillin capsules, amoxicillin and clavulanate potassium combination tablets, doxycycline, and ciprofloxacin tablets are the most widely prescribed antibiotics in Ethiopia for the control of common bacterial disease causing species [11].

Despite the considerable use of the antibiotics selected for this study (amoxicillin capsules, amoxicillin and clavulanate potassium tablets, doxycycline, and ciprofloxacin tablets) in

Ethiopia, there have not been enough studies conducted on their overall quality. Thus, this study assesses the presence of poor-quality medicines, which is very important, especially at the border of entry in Setit Humera and West Gondar Zones, where the smuggling of such products is suspected to be very high. These localities were also recently associated with conflicts and instability, one of the driving and aggravating factors for the presence and penetration of substandard and falsified (SF) medicines [12].

## Materials and methods

### General

The quality assessment of antibiotics in terms of the percentage of active pharmaceutical ingredient (API) composition by performing assay content, dissolution, hardness, friability, disintegration, and weight variation can be investigated using many standard methods [5]. Thus, the presence of poor-quality drugs can be detected by different methods, which include inspection, colorimetric methods, chromatography, and spectrometry [13].

### Description of the study area and study period

The sample collection for the study was conducted in *Setit Humera*, *West Gondar* Zones, located in North-West Ethiopia, from August to September 2021. The sample sites for medicine collection include *Humera*, *Mai Kadra*, *Metema*, *Gendawuha*, *Kokit, and Midregenet towns*. *Setit Humera* is a zone bordering Sudan and Eritrea. The West Gondar Zone borders Sudan. Because of their geographical location, both *Setit Humera* and *West Gondar* have the potential for medicine smuggling and illegal border trade.

### Sampling procedure

During sample collection in the two zones, there were around 104 health facilities (Fig 1).

Among them; samples were collected from 21 based on the purposive, convenience sampling strategy (Fig 2).

The distribution of the health facilities based on their hierarchy in the Ethiopian health system context is as follows: 47 were public health facilities (7 hospitals: 3 from *West Gondar* zones, 4 from *Setit Humera* zones), The other 40 were health centers (18 from *West Gondar*

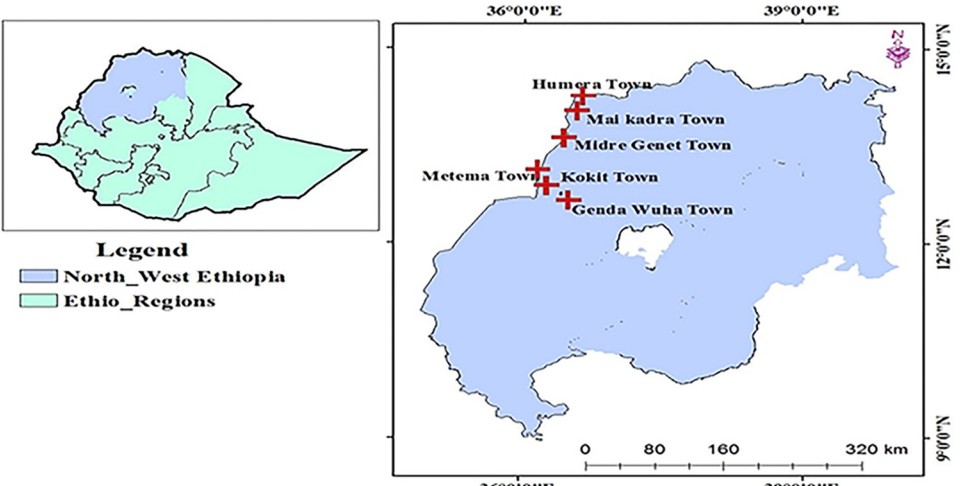

**Fig 1. Total number of health facilities that were available in the two zones during data collection.**

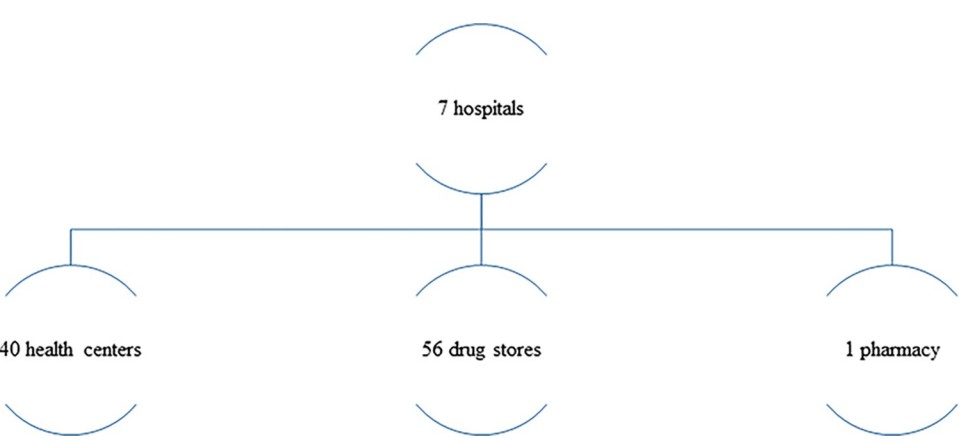

**Fig 2. Health facilities that were included in the data collection.**

zones, 22 from *Setit Humera* zones). The rest 57 were private drug retail outlets (1 pharmacy, 34 drug stores from West Gondar zones, and 22 drug stores from Setit Humera zones). Both of them were potential sites for sample collection. However, based on the sampling strategy employed, samples were collected from 21 of the health facilities (Fig 2). The sites where samples had been collected were shown on the map (Fig 3).

## Experimental

**Materials/Reagents/Chemicals.** The following instruments were used in the experiments: HPLC (Shimadzu, Japan), column (25 cm × 4.6 mm) packed with base-deactivated octadecyl-silyl silica gel for chromatography (5 μm) (Nucleosil 120-C18 and LiChrospher 100 RP 18, Germany), Ultrasonic cleaner (S. No 0083311134PO04, Dahan Scientific Co., Ltd, Korea), $p^H$ meter (S. No 31092, Biby scientific Ltd. Co., UK), shaker (S. No 100077495220240V, Germany), UV-VIS detector (S. No. L20145302582AE, Shimadzu Corporation, Japan), UV- VIS spectrophotometer (S. No. 50038, UK), Disintegration apparatus (model DIST, S.No 241002), Erweka dissolution tester apparatus (S. No. 135658–2217, Germany), Caleva hardness tester (model THT2, S.No. 111436), Erweka friability tester (S. No. 101821, Germany). Pharmaceutical primary reference standards USP amoxicillin RS (Lot No. R106H0, Spain with a potency of 86.9%), USP clavulanate lithium RS (Lot No. R071W0, India with a potency of 96.9%), USP

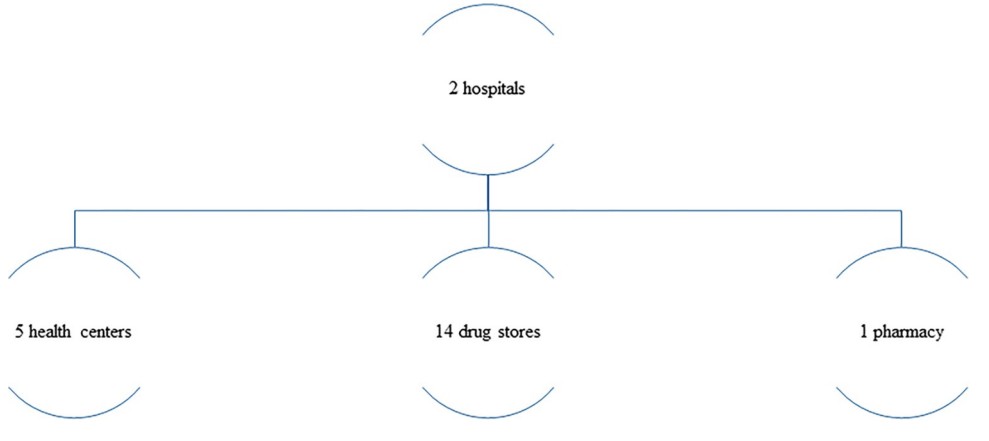

**Fig 3. Map of the study area.**

ciprofloxacin RS (Lot No. R05170, Germany with a potency of 98%), USP doxycycline RS (Lot No. R065H0, Portugal with a potency of 98%), HPLC grade methanol (B. No. MH209, made in India) and glacial acetic acid (B. No. STBG7804, made in India) were used.

**Sample collection.** Purposive sampling method was used to obtain the samples that were included in the survey. All samples were purchased from governmental and private drug retail outlets at six sites in both zones. From all sites, a total of 71 samples of amoxicillin capsules, amoxicillin and clavulanate potassium combination tablets, ciprofloxacin tablets, and doxycycline capsules were collected. Out of 71 samples, 26 were amoxicillin capsules, 12 were amoxicillin and clavulanate potassium combination tablets, 23 were ciprofloxacin tablets, and 10 were doxycycline capsules. The detailed information on the samples collected for the study is presented in S1 Table in S1 Text. Prescription papers were obtained from the University of Gondar Hospital, and a mystery shopping technique was used in the collection of the samples.

## Quality test methods

**Visual inspection.** Before sample analysis, the physical characteristics of the dosage form, packaging, and labeling information were examined for this study. The WHO/FIP/ USP joint checklist was used [7, 14]. All samples were inspected visually for a trade name, active ingredient name, manufacturer's name, and its full address. The registration status of each collected sample was checked first using the registration database, which is the Ethiopian Food and Drug Authority's (EFDA's) Electronic Regulatory Information System (ERIS) [7, 14].

**Identification test.** Identification of amoxicillin capsules was performed according to BP recommendations [15] by infrared absorption (IR). The identification of six amoxicillin and clavulanate potassium combinations and all ciprofloxacin samples was performed by HPLC by comparing the retention time of the principal peaks in the chromatogram of the samples with that of the reference primary standard [16, 17]. But for the other six amoxicillin and clavulanate potassium combination samples, tests were done by thin layer chromatography (TLC). The retardation factors of the samples and references were, respectively, compared. Identification of doxycycline capsules was performed by chemical reaction (weighing 5 mg of the contents of the capsules powder and then adding 10 mL of sulphuric acid).

**Assay.** Assay was performed by HPLC for all investigated medicines per the respective monographs of USP 2020 and BP 2019. The contents of the API of all amoxicillin and doxycycline capsules were determined using HPLC according to BP specifications, while for amoxicillin and clavulanate potassium and ciprofloxacin tablet samples, using HPLC, it was performed according to both BP 2019 and USP 2020 specifications based on their label claims.

**Dissolution test.** Dissolution of the dosages was quantified by UV spectroscopy for the API content of all investigated medicines except amoxicillin and clavulanate potassium, which were quantified by HPLC. For the UV result, the content in mg of each dissolved element was calculated by the formula:

$$\% \text{ Release} = \frac{\text{Absorbance of sample} \times \text{concentration of reference standard} \times \text{sample dilution factor} \times 100}{\text{The absorbance of reference standard} \times \text{lebel claim of sample}} \quad \text{Eq 1}$$

The dissolutions of amoxicillin and clavulanate potassium samples were evaluated using dissolution apparatus type 2 (paddle apparatus) per the United States Pharmacopoeia protocol [16]. For HPLC result, the content in mg of $C_{16}H_{19}N_3O_5S$ and $C_8H_9NO_5$ dissolved was calculated by the formula:

**Calculate the percentage of $C_{16}H_{19}N_3O_5S$ in each tablet taken.**

$$\textbf{Result} = \frac{ru}{rs} \times \frac{Cs}{Cu} \times P \times F \times 100 \qquad\qquad \text{Eq 2}$$

ru = peak response of amoxicillin from the sample solution

 rs = peak response of amoxicillin from the standard solution

 $C_S$ = concentration of USP amoxicillin RS in the standard solution (mg/mL).

 $C_U$ = nominal concentration of amoxicillin in the sample solution (mg/mL) clavulanic acid in USP clavulanate lithium RS (mg/mg)

 P = potency of USP amoxicillin RS (µg/mg)

 F = conversion factor, 0.001 mg/µg

## Calculate the percentage of $C_8H_9NO_5$ in each tablet taken

$$\textbf{Result} = \left(\frac{ru}{rs}\right) \times \left(\frac{Cs}{Cu}\right) \times P \times 100 \qquad\qquad \text{Eq 3}$$

ru = peak response of clavulanic acid from the sample solution

 rs = peak response of clavulanic acid from the standard solution

 $C_S$ = concentration of USP clavulanate lithium RS in the standard solution (mg/mL)

 $C_U$ = nominal concentration of clavulanic acid in the sample solution (mg/mL)

 P = potency of clavulanic acid in USP clavulanate lithium RS (mg/mg)

## Limits for compliance

According to the United States Pharmacopoeia 2020 (% of declared content should be): amoxicillin (92.5–110 for assay, 80 for dissolution); amoxicillin and clavulanate potassium (90–120 for assay, 85/80 for dissolution); ciprofloxacin (90–110 for assay, 80 for dissolution). British Pharmacopoeia 2019 (% of declared content): amoxicillin and clavulanate potassium (90–105 for assay, 85–80 for dissolution), and doxycycline (95–105 for assay, 85 for dissolution). All methods were validated according to USP and BP instructions for system suitability. Construction of the calibration curve line was also employed for analytical purposes.

## Dosage uniformity

The uniformity of dosage units of amoxicillin, amoxicillin-clavulanate potassium combination, ciprofloxacin, and doxycycline was demonstrated by weight variation as their active ingredient content was greater than 25 mg [16]. Dosage uniformity tests were performed according to the monographs of the USP 2020 method specifications. The requirements for dosage uniformity are met if the acceptance value of the first 10 dosage units is less than or equal to L1%, which is 15%. Then the content (assay) of the individual capsule (% Xi) was obtained by the formula:

$$\% \ Xi = \frac{\text{weight of capsule}}{\text{The average weight of 10 capsules}} \times A \qquad\qquad \text{Eq 4}$$

Were

 % Xi is the content percentage of each tablet/capsule

 A is a total percent assay that is obtained from an assay conducted on representative drug samples of each brand as indicated on the monograph for amoxicillin.

 **Disintegration time.** uncoated tablets should have a disintegration time limit of up to 15 minutes, whereas film-coated tablets should have a disintegration time limit of up to 30 minutes [16]. The average disintegration time for each product was calculated.

**Friability test.** According to the USP, the weight loss should not be more than 1%. The percentage of weight loss was calculated as follows [16].

$$\% \text{ friability} = \frac{\text{initial weight} - \text{final weight}}{\text{initial weight}} \times 100 \qquad \text{Eq 5}$$

**Hardness test.** The tester was set to crush each tablet, and the force required to crush each tablet was measured in Newton's [18].

## Data quality control and quality assurance

The data obtained were checked for completeness and consistency. The quality of the experimental results was ensured by performing system suitability tests, single injections, blank solutions, standard quality controls, checking method linearity (calibration curve), repeatability, and strictly following the procedures outlined in the compendia specification monographs. Moreover, the laboratory work was performed at an ISO 17025 qualified laboratory, following strictly the SOPs in place.

## Data management and analysis

The coded data was entered into a Microsoft Excel 2016 worksheet. It was used as it is the commonly available software. Then it was analyzed using Microsoft Excel and Origin Pro 8.5. Descriptive statistics were used to summarize the data. The mean, standard deviation, and relative standard deviation (RSD) were calculated and employed to present the data.

## Ethical consideration

Prior to sample collection, the ethical clearance approval letter (Ref. No. ERB/SOP/324/13/2021) was obtained from Addis Ababa University's College of Health Sciences, School of Pharmacy.

# Results and discussion

## Visual inspection

The results of the visual inspection were obtained after inspecting each sample physically using a visual inspection checklist. From the total 71 samples, 7.04% (5/71) were intact, and in the case of capsules, they were not easy to uncap; 1.41% (1/71) samples had problems with the packaging; 8.45% (6/71) samples labels were not legible; 50.70% (36/71) samples did not have the symbol ® following the trade name; and 7.04% (5/71) samples had not mentioned the manufacturer's name on their label. Overall, 12.07% (17/71) of the observed samples did not comply with the criteria given in the joint WHO/FIP/USP checklists. Trade name declarations without properly registering and obtaining market rights with the symbol "®" are not a standard way of practice. It may be practiced to infringe on another company's' registered trademarks. Similar results have been seen in studies conducted in Rwanda [7] and Jimma, Ethiopia [5]. This study indicates the presence of a loophole in the medicine regulation system of the country, resulting in the medicines that did not meet the official specifications for packaging or labeling reaching the country's medicine consumers. Hence, it had to strengthen the regulation system, especially in the prone areas where the study sites are located.

## Registration status of medicine

From the total 71 samples included in this study, 42 (59.15%) samples were registered, 18 (25.35%) were not registered, and 11 (15.49%) were registered in the earlier years, but the

licenses expired during the study period and need re-licensure of registration by the regulatory authority, i.e., by EFDA. Of the total collected samples, 53.52% (38/71) were manufactured outside Ethiopia, 7.89% (3/38) of the imported samples were not registered (but the study did not consider donated medicines that may enter without the full registration process documentation in place and those that underwent continuous regulatory processes after the study period), and 46.47% (33/71) of the collected samples were manufactured in Ethiopia, and from this, 45.45% (15/33) were not registered. 21.21% (7/33) of the samples were manufactured in Ethiopia and registered, but the license period had expired. Similarly, 10.52% (4/38) of the samples were manufactured outside Ethiopia and were registered, but the license period had already expired.

## Identification test

As indicated in S1a and S1b Fig in S1 Text, the IR absorption spectra of the standard and amoxicillin samples analyzed in this study were similar. The peak retention times of the amoxicillin and clavulanate potassium combination selected samples were found to be similar to the retention time of the HPLC chromatogram of amoxicillin and clavulanate potassium USP RS. Similarly, the retention times of the peaks in the chromatograms of all ciprofloxacin samples were similar to their respective RS. The tested products' identity was confirmed by the chemical reaction results, which means that the required yellow color was produced. As a result, all tested samples (71) passed the identification test. Similar results were seen in studies conducted in different areas of the world [4, 5, 19–24] including in studies done in Sub-Saharan Africa.

## Assay test

The assay values for amoxicillin drug products were found to meet the pharmacopeial requirement (BP 92.5 to 110%), ranging from 97.22 to 105% (S2 Table in S1 Text). However, 50% (6/12) of amoxicillin and clavulanate potassium samples (S3 in S1 Text), 47.8% (11/23) of ciprofloxacin samples (S4 Table in S1 Text), and 60% (6/10) of doxycycline samples (S5 Table in S1 Text) did not meet the specification for the assay and thus are formally classified as substandard medicines.

## Dissolution

All samples of amoxicillin, ciprofloxacin, and doxycycline released the expected amount of API within time and met the pharmacopeial tolerance limits. However, all amoxicillin and clavulanate potassium combination brands did not meet the pharmacopeial tolerance limits. Different factors might be attributable to the failure, including api solubility, polymorphism, and the different excipients used in the formulation. The dissolution profiles obtained for the different brands are shown in Figs 4–8.

## Weight variation

For uniformity of dosage units, all amoxicillin samples were in line with pharmacopeial acceptance criteria for dosage uniformity. However, 25% (3/12) of the amoxicillin and clavulanate potassium samples, 17.39% (4/23) of the ciprofloxacin samples, and 30% (3/10) of the doxycycline samples did not meet these specifications. It might be due to the powder flow property, the machine systems used in filling capsules or for compression purposes in tablets, and also other unit operation processes involved in the formulation. Besides, the use of a master formula that is not properly validated and optimized, as well as the use of improper lubricants;

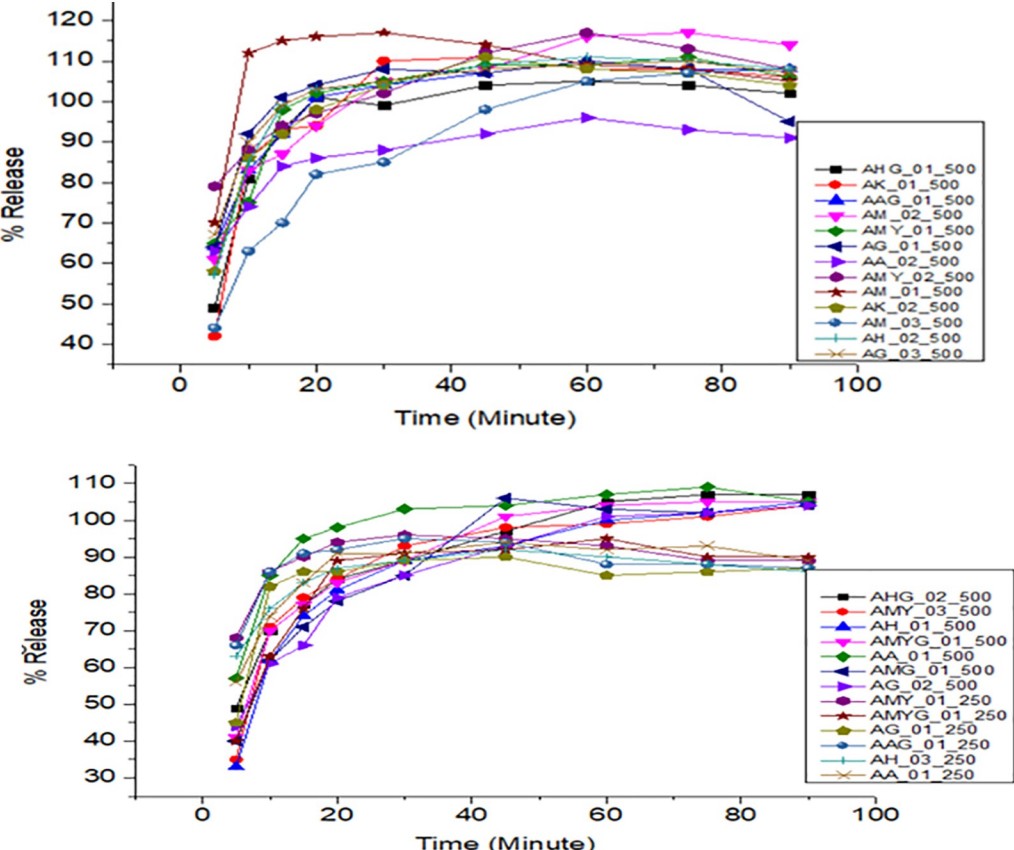

**Fig 4. Time-dependent dissolution profiles of different samples of amoxicillin.**

might be the root cause of the failure. For details about the data, please look at S6-S9 Tables in S1 Text.

## Disintegration test

For the disintegration test, all batches of amoxicillin, ciprofloxacin, and doxycycline disintegrated in less than 15 minutes, i.e., complied with the specification limit. However, two batches of amoxicillin and clavulanate potassium combinations failed to comply with the USP specification. It might be due to different contributing factors in harmony or alone, which include the concentration and type of dis-integrant(s) used, coating and additional procedures, type and nature of excipients, as well as the API(s) chemical and physical nature. Detailed data are presented in the S7-S9 Tables in S1 Text.

## Friability test

For the friability test, all batches of ciprofloxacin, amoxicillin, and clavulanate potassium complied with USP specifications. But three batches of ciprofloxacin 500 mg tablets (CHG_02_500, CMY_01_500, and CM_03_500) were broken during the friability test, i.e., failed the test. The cause of the failure might be the concentration of the binder added, the crushing strength used in the formulation, and other related processes. Data is given in S7 and S8 Tables in S1 Text

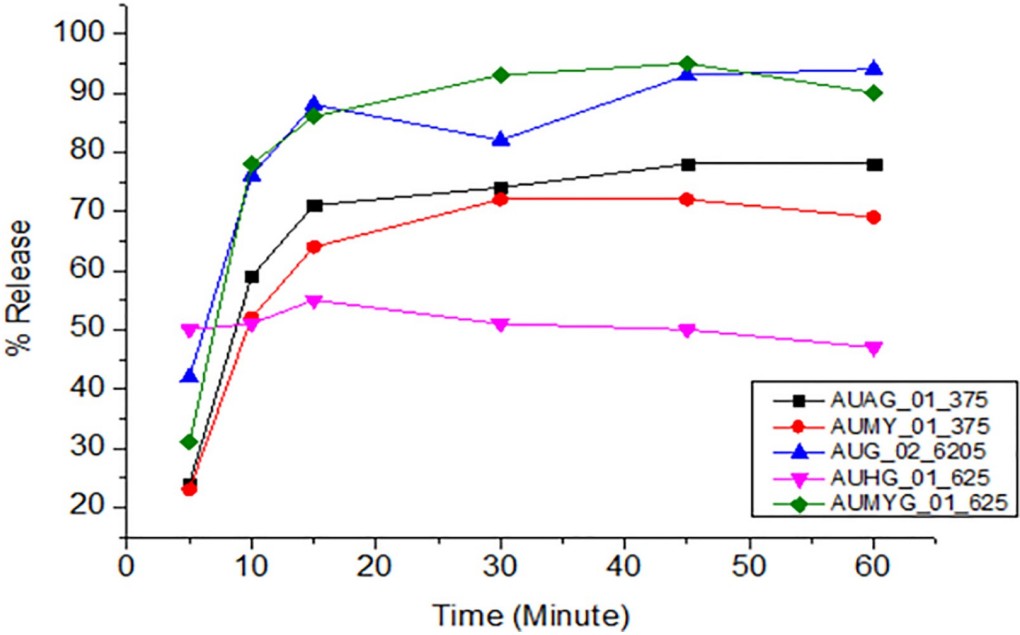

**Fig 5. Time-dependent-dissolution profiles of amoxicillin in amoxicillin and clavulanic acid tablets.**

## Hardness test

All ciprofloxacin samples, amoxicillin, and clavulanate potassium tablets examined gave a hardness value > 40 N. Thus, all products conformed to the requirement for the hardness test. Data is given in S7 and S8 Tables in S1 Text.

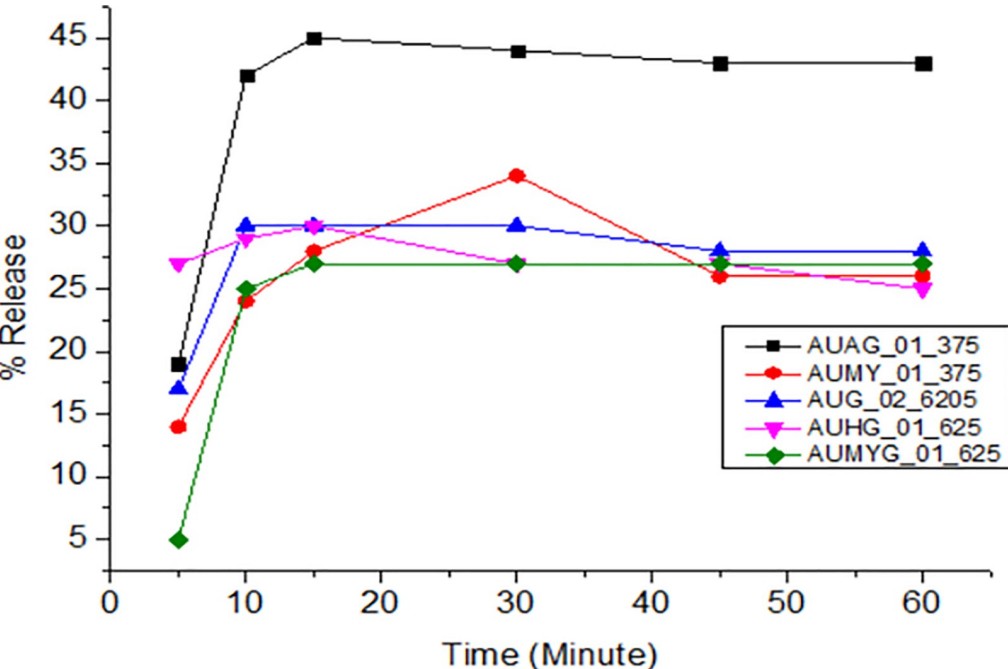

**Fig 6. Time-dependent-dissolution profiles of clavulanic acid in amoxicillin and clavulanic acid tablets.**

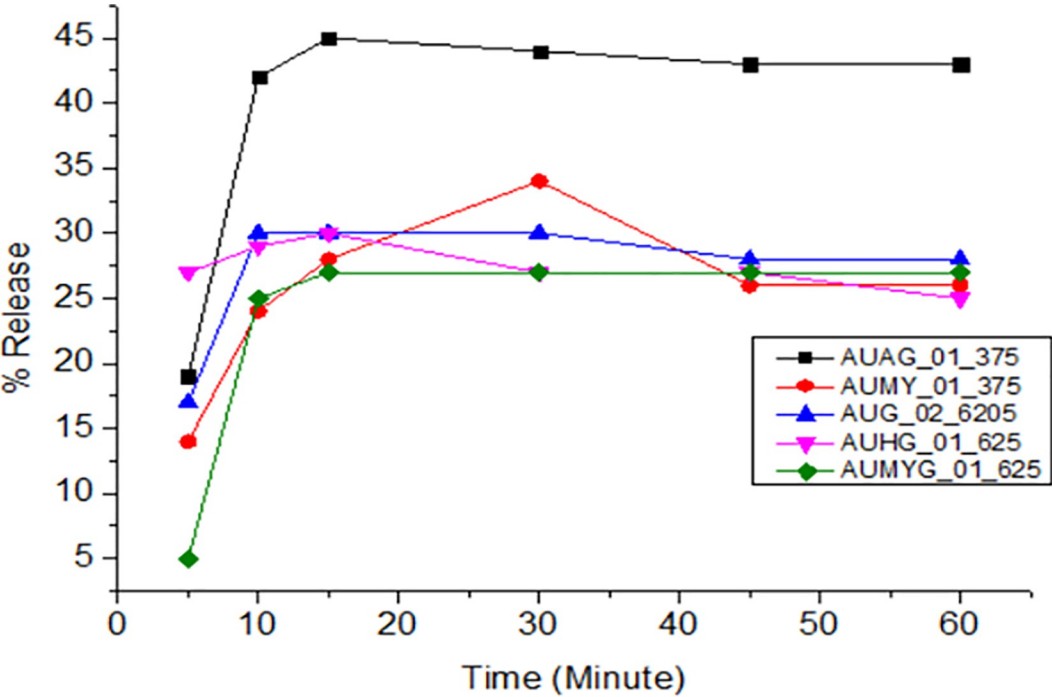

**Fig 7. Time-dependent-dissolution profiles of different samples of ciprofloxacin.**

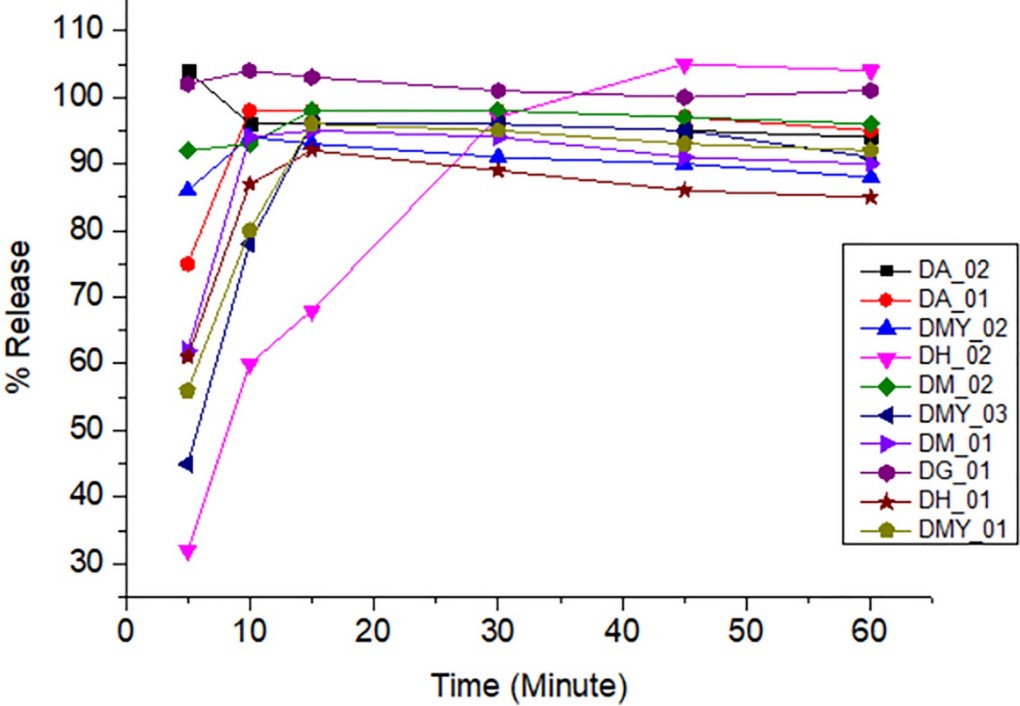

**Fig 8. Time-dependent-dissolution profiles of doxycycline.**

## Assay limits for system suitability parameters

Theoretical plate values for amoxicillin capsules were 2779, 7957.88 for amoxicillin tablets, 7114.75 for clavulanate potassium tablets by USP, and 8339 for amoxicillin tablets and 7276 for clavulanate potassium tablets by BP, respectively. Doxycycline's theoretical plate value was 2323. By USP and BP methods, the resolution factor for amoxicillin capsules was 3.6, 9.9 for amoxicillin and clavulanate potassium tablets, and 9.5 and 7.6 for ciprofloxacin tablets. Tailing factors for amoxicillin capsules were 1, 0.96, and 1.44 for amoxicillin and clavulanate potassium tablets, respectively; 1.3 and 1.5 for ciprofloxacin tablets by USP and BP methods; and 1 for doxycycline capsules. The relative standard deviation for amoxicillin capsules was 0.039%, (0.12% for amoxicillin, 0.1% for clavulanate potassium by USP) and (0.15% for amoxicillin, 0.1% for clavulanate potassium by BP) for amoxicillin and clavulanic acid tablets, (0.2%, 1.5%) for ciprofloxacin by USP and BP methods, respectively, and 0.24% for doxycycline capsules.

The dissolution profiles of all brands of amoxicillin and clavulanate potassium did not comply with the USP 2020 and BP-2019 specification limits. The dissolution profile of amoxicillin and clavulanic acid tablets was illustrated in Figs 5 and 6.

As shown in Fig 7, all samples of ciprofloxacin showed more than 80% drug release after 30 minutes. At 30 minutes, all the samples released more than the pharmacopeia requirement of 80% active pharmaceutical ingredient [16].

The average values of dissolution of the different samples of doxycycline tested are shown in Fig 8. The obtained dissolution content at 30 minutes was found ranging from 94.81% ± 0.90% to 101.68% ± 0.91%. All samples of doxycycline showed more than 80% drug release after 30 minutes. At 30 minutes, all the samples released more than the pharmacopeia requirement of 80% active pharmaceutical ingredient.

The results of this study revealed that a significant amount of substandard antibacterial medicines were marketed in Ethiopia-Sudan-Eritrea corridor. Samples were failed for uniformity of dosage forms, dissolution, assay, disintegration, hardness, and friability, as shown in Table 1. Similar results were seen in a study conducted in Jimma, Ethiopia where failed doxycycline samples were seen for physical tests. In addition in a study conducted in Nigeria's Katsina State [19], where 30.0% of ciprofloxacin did not meet the specifications for assay and in Ghana [22] 100% failure (14 out of 14) for amoxicillin and 14% failure (2 out of 14) for clavulanate potassium. In a study that was conducted in Jounieh, Lebanon [23], It was reported that all ciprofloxacin samples analyzed failed the assay test. The results of this study showed that 40% of doxycycline samples comply with the BP specification. However, 60% of the samples failed the assay test. Of the total failed samples, five of them were lower than 95% (53.6–75.4%) and for a sample, the API content was higher than 105% of the label claim.

**Table 1. Summary of quality control test results of products (% failure (n = number of failed samples out of total n samples analyzed)).**

| Product | Identification | Dissolution | Uniformity of Dosage forms | Assay | Disintegration | Friability | Hardness |
|---|---|---|---|---|---|---|---|
| **Amoxicillin** | 0% (0/26) | 0% (0/26) | 0% (0/26) | 0% (0/26) | 0% (0/26) | 0% (0/26) | 0% (0/26) |
| **amoxicillin and clavulanate potassium combination** | 0% (0/12) | 100 (12/12) | 25% (3/12) | 50% (6/12) | (16.7%) 2/12 | 0% (0/12) | 0% (0/12) |
| **Ciprofloxacin** | 0% (0/23) | 0% (0/23) | 17.39% (4/23) | 47.8% (11/23) | 0% (0/23) | (13.04%)3/23 | 0% (0/23) |
| **Doxycycline** | 0% (0/10) | 0% (0/10) | 30% (3/10) | 60% (6/10) | 0% (0/10) | 0% (0/10) | 0% (0/10) |

The dissolution profiles of amoxicillin samples were shown in Fig 4. All amoxicillin samples analyzed had complied with the pharmacopoeial specifications.

## Limitations

The study's findings could have been influenced by the following limitations. To begin, only capsules and tablets were tested. As a result, the findings are inapplicable to other dosage forms or medications. Second, because of the small number of samples collected and the method of sample collection is based on convenience sampling technique which may be implied in poor representativeness, the study findings might not represent the entire country.

## Conclusion

Poor quality medicines were still found on the market despite the Ethiopian Food and Drug Authority(EFDA's) i.e. the national medicine regulatory agency ongoing efforts to ensure that medications circulating on the market are of high quality, safe, and effective through pre-registration assessment, product quality screening at ports of entry, and routine product inspection and surveillance. Nevertheless, falsified medicine that is without any API, was not encountered during the assessment. Although appropriate regulatory actions have been taken, such as recalling substandard batches and removing falsified medicines from the market, this study has revealed that the importance of continuing and strategically implementing post-marketing surveillance programs as one of Ethiopia's key regulatory functions. In this study as some of the medicines analyzed did not fulfill the pharmacopoeial requirements, therefore, even if the area was under the conflict situation at the time of this study, the regulatory body should focus on enforcing the necessary measures by collaborating with the necessary stake holders like regional regulatory body and others to ensure that antibacterial drugs on the market meet the required mandatory minimum standards.

## Supporting information

**S1 Text. Supplementary file that contain additional results and data.**
(DOCX)

## Acknowledgments

We want to thank EFDA for their permission to do the work in their medicine quality control laboratory. University of Gondar kindly offered salaried study leave for the MSc study to TD. United states Pharmacopoeia (USP) kindly offered us the reference standards used in the study through its academic support program to AA.

## Author Contributions

**Conceptualization:** Ayenew Ashenef.

**Data curation:** Tewodros Denekew, Ayenew Ashenef.

**Formal analysis:** Tewodros Denekew, Yehualashet Teshome, Ayenew Ashenef.

**Funding acquisition:** Ayenew Ashenef.

**Investigation:** Tewodros Denekew, Siyalkalem Endeshaw, Ayenew Ashenef.

**Methodology:** Siyalkalem Endeshaw, Ayenew Ashenef.

**Project administration:** Ayenew Ashenef.

**Resources:** Ayenew Ashenef.

**Supervision:** Tadele Eticha, Ayenew Ashenef.

**Validation:** Ayenew Ashenef.

**Writing – original draft:** Ayenew Ashenef.

**Writing – review & editing:** Tadele Eticha, Ayenew Ashenef.

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
