## [Editor Report · Decision Letter 0]

6 Nov 2023

PONE-D-23-33682

Post-marketing Quality Surveillance of Selected Antibacterial Agents Marketed in Porous Borders; the Case of Ethiopian side of Ethiopia-Sudan-Eritrea corridor.

PLOS ONE

Dear Dr. Ashenef,

Thank you for submitting your manuscript to PLOS ONE. After careful consideration, we have decided that your manuscript does not meet our criteria for publication and must therefore be rejected.

Specifically:

1-Please provide rational of selecting the sample sites. How and why you select the sites? Detail methodology is mandatory.

2-Inside experimental, Please write two headings 1) Instrumentation 2) Chemicals. Provide the source of all items and % purity of chemical used. Please provide details on the validation of instrument. Yes or No. If Yes then who validated the instruments? line 129-144 has been written poorly and several source missing. Kindly read quality manuscript on how to write the details. The source details are mandatory. The country name is not enough to justify.

3- The study design is missing such as sampling methodology, sample size calculation. The current methodology need to improve before we procced. Include the heading of study design and explain each items.

4- The experimental details with reference were missing such as Assay methods.

5- There is no rational on the selection of the United States Pharmacopoeia 2020? Why? It is necessary to provide the details. There is no sampling methodology, we do not know the product follow which standard (BP/USP). We cannot generalized the results on one standard. 

I am sorry that we cannot be more positive on this occasion, but hope that you appreciate the reasons for this decision.

Kind regards,

Muhammad Shahzad Aslam, Ph.D.,M.Phil., Pharm-D

Academic Editor

PLOS ONE

Additional Editor Comments:

1-Please provide rational of selecting the sample sites. How and why you select the sites? Detail methodology is mandatory.

2-Inside experimental, Please write two headings 1) Instrumentation 2) Chemicals. Provide the source of all items and % purity of chemical used. Please provide details on the validation of instrument. Yes or No. If Yes then who validated the instruments? line 129-144 has been written poorly and several source missing. Kindly read quality manuscript on how to write the details. The source details are mandatory. The country name is not enough to justify.

3- The study design is missing such as sampling methodology, sample size calculation. The current methodology need to improve before we procced. Include the heading of study design and explain each items.

4- The experimental details with reference were missing such as Assay methods.

5- There is no rational on the selection of the United States Pharmacopoeia 2020? Why? It is necessary to provide the details. There is no sampling methodology, we do not know the product follow which standard (BP/USP). We cannot generalized the results on one standard.

- - - - -

---

## [Author Response · Author response to Decision Letter 0]

8 Dec 2023

November 16,2023

Response to the decision Letter:

1-Please provide rationale for selecting the sample sites. How and why you select the sites? Detail methodology is mandatory.

Ans: The sites were selected as potential areas of cross -border trade that includes medicines that enter without proper regulatory (National medicines Authority) oversight. This statement hd been included in the revision ( Please look to line 115-117) 

2-Inside experimental, Please write two headings 1) Instrumentation 2) Chemicals. Provide the source of all items and % purity of chemical used. Please provide details on the validation of instrument. Yes or No. If Yes then who validated the instruments? line 129-144 has been written poorly and several source missing. Kindly read quality manuscript on how to write the details. The source details are mandatory. The country name is not enough to justify.

Ans: Instrumentation, chemicals with two heading and the necessary details had been included easily in the revision. (Look at lines 125-140)

We had used the national medicines authority laboratory an ISO 17025 accredited ones and these are routine procedures they abide for compliance and the details of who and when can be easily provided in the manuscript if needed. The sentences mentioning this fact had been included in the revision (line126-128)

3- The study design is missing such as sampling methodology, sample size calculation. The current methodology need to improve before we procceed. Include the heading of study design and explain each items.

Ans: The study design followed WHO recommendations for post marketing quality surveillance protocol and this had been enumerated in the revision easily. (line 126-128)

4- The experimental details with reference were missing such as Assay methods.

 Ans: It is either USP/BP based on the specific medicine label claim and this can be easily incorporated in the revisions.

5- There is no rational on the selection of the United States Pharmacopoeia 2020? Why? It is necessary to provide the details. There is no sampling methodology, we do not know the product follows which standard (BP/USP). We cannot generalized the results on one standard.

Ans: Standards were selected based on the Medicines label claim of compliance and all were not analyzed in one standard. We had used BP/USP standards accordingly. we can easily clarify it on the revisions.

Thank you for the important critical inputs and scientific justifications 

sincerely,

Ayenew Ashenef ( on behalf of the authors)

---

## [Decision Letter · Decision Letter 1]

21 Mar 2024

PONE-D-23-33682R1Post-marketing Quality Surveillance of Selected Antibacterial Agents Marketed in Porous Borders; the Case of Ethiopian side of Ethiopia-Sudan-Eritrea corridor.PLOS ONE

Dear Dr. Ayenew Ashenef,

Thank you for submitting your manuscript to PLOS ONE. After careful consideration, we feel that it has merit but does not fully meet PLOS ONE’s publication criteria as it currently stands. Therefore, we invite you to submit a revised version of the manuscript that addresses the points raised during the review process.

We look forward to receiving your revised manuscript.

Kind regards,

Obed Kwabena Offe Amponsah, PharmD, Ph.D.

Academic Editor

PLOS ONE

1. Please ensure that your manuscript meets PLOS ONE's style requirements, including those for file naming. The PLOS ONE style templates can be found at https://journals.plos.org/plosone/s/file?id=wjVg/PLOSOne_formatting_sample_main_body.pdf and https://journals.plos.org/plosone/s/file?id=ba62/PLOSOne_formatting_sample_title_authors_affiliations.pdf.

“We want to thank EFDA for their permission to do the work in their medicine quality control laboratory. Addis Ababa University graduate thesis work support to TD and the Ministry of Innovation and Technology, government of Ethiopia to AA funded this study.”

‘Addis Ababa University to TD Ministry of Innovation and technology, government of Ethiopia to AA.”

4. Please remove your figures from within your manuscript file, leaving only the individual TIFF/EPS image files, uploaded separately. These will be automatically included in the reviewers’ PDF.

Reviewers' comments:

Reviewer's Responses to Questions

**Comments to the Author**

1. If the authors have adequately addressed your comments raised in a previous round of review and you feel that this manuscript is now acceptable for publication, you may indicate that here to bypass the “Comments to the Author” section, enter your conflict of interest statement in the “Confidential to Editor” section, and submit your "Accept" recommendation.

Reviewer #1: (No Response)

Reviewer #2: (No Response)

2. Is the manuscript technically sound, and do the data support the conclusions?

Reviewer #1: Partly

Reviewer #2: Yes

3. Has the statistical analysis been performed appropriately and rigorously? 

Reviewer #1: (No Response)

Reviewer #2: N/A

4. Have the authors made all data underlying the findings in their manuscript fully available?

Reviewer #1: (No Response)

Reviewer #2: Yes

5. Is the manuscript presented in an intelligible fashion and written in standard English?

Reviewer #1: Yes

Reviewer #2: Yes

6. Review Comments to the Author

Reviewer #1: Reviewer comments (PONE-D-23-33682R1)

General comments

• The Authors are appreciated for using an ISO 17025 qualified laboratory for conducting the quality test.

• There are grammatical errors and punctuation marks that need to be addressed throughout the manuscript and I recommend proof-reading of the manuscript again.

• Use terms consistently throughout the manuscript. For example the words medicine Vs drug, falsified Vs counterfeit. Better to remove the word counterfeit from the article.

• Contesting/disputing issues such as locations, administrative demarcation on the study area, or use of unofficial demarcation of territories in scientific studies is not encouraged. Better to be silent or indicate in neutral way.

• The National Regulatory Authority in Ethiopia has a guideline for PMS including data collection; did you use that or any other model for your study? Describe this.

Specific comments

1. Title of the study (line 1-2)

• Delete the phrase ‘’Ethiopia side of’’.

• Replace ; by :

• Rephrased title: Post-marketing Quality Surveillance of Selected Antibacterials Marketed in Porous Borders: the Case of Ethiopia-Sudan-Eritrea border.

Abstract (lines 25-51)

• Though the abstract contains all the contents, it is good to structure it into background, objectives, Methods, Results and Conclusions.

• Better to replace the phrase ‘low-quality medicines’ to poor-quality medicines’

• Line 31, delete corridor

• Lines 35-36, use the correct name (i.e. replace electronic registration Information system(ERIS) database by Electronic Regulatory Information System (eRIS).

• Line 38 reads as 15.49% (11/71) were registered but the license period had expired. Have you checked the current application status in the eRIS? This comment also applies to the result section.

• Line 47, do you think the conclusion is appropriate. During the study period, there was war/conflict in the area which might be open for illegal distribution of medicines. It is wise to adjust the conclusion considering the situation.

 

Introduction

• The introduction has coherence problem. Some concepts that are not aligned with this study (example disease burden) should be removed. The justification should also be stated in one paragraph. It should not be stated here and there (lines 84-85, line 97…, line 99 ,,,, etc.

• Lines 72-73, proper regulation includes both continuous post-marketing surveillance, and fast regulatory actions. Either delete both continuous post-marketing surveillance and fast regulatory actions or focus on ‘post-marketing surveillance’.

• Lines 75-77 states about the burden of infectious disease. This is another concept which does not align with the research objective. You can mention it as a problem that can be exacerbated by substandard and falsified products.

• Lines 81-85; care should be taken here. 1). Are you sure about the absence of comprehensive evaluation of basic regulatory functions, 2). I do not think, it is worthy to mention about maturity level in this study. Better to remove lines 81-83. Line 84 and 85 are enough to justify for your study.

• Line 87, replace with national drug policy or use the correct name.

• Align lines 81-82, and the last paragraph of the introduction section.

• Lines 90-94, better to indicate in the method section

• Seriously review lines 98-104 - repetition of terms and concepts.

• Line 107 is repetition. It is stated earlier in the introduction section

Methods and Materials

• Properly describe the method section

• Line 117: Both zones are located in the Amhara regional state of Ethiopia. This is a contesting issue. Better to to remove.

• From the 114 health institutions available, how many of them did you include in the data collection process? Please, make it clear.

• What is your sample size and how do you calculate it? The authors should clarify and describe it.

• Sampling procedure is not clearly outlined. How do you select your sampling sites from the existing sites? In addition, describe your inclusion and exclusion criteria for both the medicines to be collected and the sampling sites?

• The authors should describe the sampling technique.

• Line 160, correct it as commented earlier.

• Line 230: describe why the authors use excel sheet for the data management and analysis?

Results and discussion

• Line 257: do you think, the reason is weak regulation during release at the port? The illegal medicines might enter to the country through other channels which were unknown to the regulatory authority. Instead of using the word release, it is better to use enter or crossing the border.

• Line 259: It is a concerning issue for the Ethiopian Medicine Regulatory Agency (MRA). Does this statement value to your results and discussion? Better to remove it.

• In your results and discussion, it is wise to remove the use of falsified medicines. The authors did not encounter falsified medicines. Example line 263.

• Lines 261-263, better remove it. It is known and practiced that if this illegal practice is known or encountered by the authority, strict actions has been taken.

• Line 268: only 7.89% (3/38) of the imported samples were not registered. What if these three products are authorized by the authority though they are not register? In Ethiopian law, there are circumstances that unregistered but authorized medicine may enter to the market of Ethiopia. Example donation. The Authors should consider the circumstances that are allowed by the Ethiopian law or you need to review the laws and reviser your results about the issue of registration of medicines. This also include for the locally produced medicines.

• For the samples failed to meet the standards in the identification, assay, disillusion, disintegration, etc., tests; what is the reason behind for the failure? It is good to indicate the possible reasons (i.e. failure of manufacturing, storage, transportation, or other reasons).

Conclusions

• Line 388: make it conclusions

• Lines 389-392: already stated in the result section. Adjust the conclusions accordingly.

• Line 393: replace the regulatory authority (EFDA) by the Ethiopian Food and Drug Authority and the region regulatory bodies

Reviewer #2: The introduction is well written and states clearly the matter at hand.

Method

The materials and methods used have been well elaborated

Results and Discussion

Visual inspection

"Hence, it had to strengthen the regulation system, especially in the prone areas where the study sites are located"

Consider changing had to has

For most of the subsections under results and discussion they appear to be only results. Please consider discussing the subheadings as well.

7. PLOS authors have the option to publish the peer review history of their article (what does this mean?). If published, this will include your full peer review and any attached files.

Reviewer #1: No

Reviewer #2: No

---

## [Author Response · Author response to Decision Letter 1]

22 May 2024

The rebuttal letter is attached. It addresses all comments and points raised by the editor and the reviewers.

---

## [Decision Letter · Decision Letter 2]

8 Jul 2024

PONE-D-23-33682R2Post-marketing Quality Surveillance of Selected Antibacterial Agents Marketed in Porous Borders; the case of Ethiopia-Sudan-Eritrea border.PLOS ONE

Dear Dr. Ashenef,

Thank you for submitting your manuscript to PLOS ONE. After careful consideration, we feel that it has merit but does not fully meet PLOS ONE’s publication criteria as it currently stands. Therefore, we invite you to submit a revised version of the manuscript that addresses the points raised during the review process.

Please see the reviewer comments on updating the references as well as checking for topographical and grammar errors in the manuscript. If possible, I would recommend using an English grammar service to address this if you desire. Please pay attention to the reviewer comments to further improve the manuscript. Thank you for the effort put into making this manuscript a more improved version of the initial submission.

We look forward to receiving your revised manuscript.

Kind regards,

Obed Kwabena Offe Amponsah, PharmD, Ph.D.

Academic Editor

PLOS ONE

Journal Requirements:

Additional Editor Comments (if provided):

Reviewers' comments:

Reviewer's Responses to Questions

**Comments to the Author**

1. If the authors have adequately addressed your comments raised in a previous round of review and you feel that this manuscript is now acceptable for publication, you may indicate that here to bypass the “Comments to the Author” section, enter your conflict of interest statement in the “Confidential to Editor” section, and submit your "Accept" recommendation.

Reviewer #1: All comments have been addressed

Reviewer #2: All comments have been addressed

2. Is the manuscript technically sound, and do the data support the conclusions?

Reviewer #1: Yes

Reviewer #2: Yes

3. Has the statistical analysis been performed appropriately and rigorously? 

Reviewer #1: Yes

Reviewer #2: N/A

4. Have the authors made all data underlying the findings in their manuscript fully available?

Reviewer #1: Yes

Reviewer #2: Yes

5. Is the manuscript presented in an intelligible fashion and written in standard English?

Reviewer #1: Yes

Reviewer #2: Yes

6. Review Comments to the Author

Reviewer #1: All comments are addressed as per the recommendation. The authors should still conduct topographic and grammatical checks. It is worth to mention one important issues to be addressed. No falsified medicine was encountered during the assessment and it is good to indicate this in the result and conclusion part of the manuscript.

Reviewer #2: The manuscript looks better now.

However, some of your references are rather too old.

Reference 1 although very old appears to support a historic fact and this is allowed.

References 8, 9 and 11 however are over 10 years old and do not accurately reflect currents status of the situations as cited in the manuscript. In most of these cases recent literature are available, consider replacing or citing alongside more recent literature.

7. PLOS authors have the option to publish the peer review history of their article (what does this mean?). If published, this will include your full peer review and any attached files.

Reviewer #1: No

Reviewer #2: No

---

## [Author Response · Author response to Decision Letter 2]

15 Jul 2024

July 12, 2024

Rebuttal letter

Dear 

Obed Kwabena Offe Amponsah, PharmD, Ph.D.

Academic Editor

PLOS ONE

First and foremost, thank you for due consideration of our manuscript PONE-D-23-33682R2

Post-marketing Quality Surveillance of Selected Antibacterial Agents Marketed in Porous Borders; the case of Ethiopia-Sudan-Eritrea border. We want to acknowledge the scientific in puts kindly offered by the anonymous reviewer’s that helped us to improve our manuscript to best scientific standards.

Below, please look at the point by point replies to the reviewer’s comments:

Comments to the Author

1. If the authors have adequately addressed your comments raised in a previous round of review and you feel that this manuscript is now acceptable for publication, you may indicate that here to bypass the “Comments to the Author” section, enter your conflict of interest statement in the “Confidential to Editor” section, and submit your "Accept" recommendation.

Reviewer #1: All comments have been addressed

Reviewer #2: All comments have been addressed

Answer: Thank you for confirming our best attempt to address the previous comments

2. Is the manuscript technically sound, and do the data support the conclusions?

Reviewer #1: Yes

Reviewer #2: Yes

Answer: Thanks

3. Has the statistical analysis been performed appropriately and rigorously? 

Reviewer #1: Yes

Reviewer #2: N/A

Answer: Thanks

4. Have the authors made all data underlying the findings in their manuscript fully available?

Reviewer #1: Yes

Reviewer #2: Yes

Answer: All data used to wrote the manuscript is included in the body of the paper as well as as supplementary files. 

5. Is the manuscript presented in an intelligible fashion and written in standard English?

Reviewer #1: Yes

Reviewer #2: Yes

Answer: The manuscript had been revised to address typographical and grammatical errors thus it fulfills the standard scientific English writing.

6. Review Comments to the Author

Reviewer #1: All comments are addressed as per the recommendation. The authors should still conduct topographic and grammatical checks. It is worth to mention one important issues to be addressed. No falsified medicine was encountered during the assessment and it is good to indicate this in the result and conclusion part of the manuscript.

Answer; Grammatical and typographical issues had been addressed extensively. Per the recommendation, the conclusion had included the above statement (Line 392)

Reviewer #2: The manuscript looks better now.

However, some of your references are rather too old.

Reference 1 although very old appears to support a historic fact and this is allowed.

References 8, 9 and 11 however are over 10 years old and do not accurately reflect currents status of the situations as cited in the manuscript. In most of these cases recent literature are available, consider replacing or citing alongside more recent literature.

Answer; Per the recommendations the above references more than 10 years old had been replaced by recent references containing similar facts as the outdated references. Please look at lines 452ff,457ff, 467ff

7. PLOS authors have the option to publish the peer review history of their article (what does this mean?). If published, this will include your full peer review and any attached files.

Do you want your identity to be public for this peer review? For information about this choice, including consent withdrawal, please see our Privacy Policy.

Reviewer #1: No

Reviewer #2: No

Answer: From the authors perspective, we declared that the reviewer’s ideas with our responses to the scientific comments in anonymized manner can be published among the different revision versions. 

With best regards,

Ayenew Ashenef,

On Behalf of the Authors

---

## [Editor Report · Decision Letter 3]

19 Jul 2024

Post-marketing Quality Surveillance of Selected Antibacterial Agents Marketed in Porous Borders; the case of Ethiopia-Sudan-Eritrea border.

PONE-D-23-33682R3

Dear Dr. Ashenef,

We’re pleased to inform you that your manuscript has been judged scientifically suitable for publication and will be formally accepted for publication once it meets all outstanding technical requirements.

Please ensure to remove the word "that" from line 392 in the conclusion during the proof stage.

Kind regards,

Obed Kwabena Offe Amponsah, PharmD, Ph.D.

Academic Editor

PLOS ONE

---

## [Editor Report · Acceptance letter]

1 Aug 2024

PONE-D-23-33682R3 

PLOS ONE

Dear Dr. Ashenef, 

I'm pleased to inform you that your manuscript has been deemed suitable for publication in PLOS ONE. Congratulations! Your manuscript is now being handed over to our production team.

Kind regards, 

on behalf of

Dr. Obed Kwabena Offe Amponsah 

Academic Editor

PLOS ONE